# (GIGA)bYte

DATA RELEASE

# Draft genome of the endangered visayan spotted deer (*Rusa alfredi)*, a Philippine endemic species

Ma. Carmel F. Javier[1], Albert C. Noblezada[1], Persie Mark Q. Sienes[2,*], Robert S. Guino-o[3], Nadia Palomar-Abesamis[2], Maria Celia D. Malay[4], Carmelo S. del Castillo[5,6] and Victor Marco Emmanuel N. Ferriols[1,5,*]

1 Philippine Genome Center Visayas, University of the Philippines Visayas, Miagao Iloilo, Philippines
2 Biology Department, Silliman University, Dumaguete, Philippines
3 Angelo King Center for Research and Environmental Management, Silliman University, Dumaguete, Philippines
4 Marine Science Institute, University of the Philippines Diliman, Quezon City, Philippines
5 Institute of Aquaculture, College of Fisheries and Ocean Sciences, University of the Philippines Visayas, Miagao Iloilo, Philippines
6 National Institute of Molecular Biology and Biotechnology, University of the Philippines Visayas, Miagao Iloilo, Philippines

## ABSTRACT

The Visayan Spotted Deer (VSD), or *Rusa alfredi*, is an endangered and endemic species in the Philippines. Despite its status, genomic information on *R. alfredi*, and the genus *Rusa* in general, is missing. This study presents the first draft genome assembly of the VSD using the Illumina short-read sequencing technology. The resulting RusAlf_1.1 assembly has a 2.52 Gb total length, with a contig N50 of 46 Kb and scaffold N50 size of 75 Mb. The assembly has a BUSCO complete score of 95.5%, demonstrating the genome's completeness, and includes the annotation of 24,531 genes. Our phylogenetic analysis based on single-copy orthologs revealed a close evolutionary relationship between *R. alfredi* and the genus *Cervus*. RusAlf_1.1 represents a significant advancement in our understanding of the VSD. It opens opportunities for further research in population genetics and evolutionary biology, potentially contributing to more effective conservation and management strategies for this endangered species.

**Submitted:** 11 October 2024

\* Corresponding authors. E-mail: persieqsienes@su.edu.ph; vnferriols@up.edu.ph

Preprint submitted at https://doi.org/10.1101/2025.02.05.636739

**Subjects** Genetics and Genomics, Bioinformatics, Evolutionary Biology

## DATA DESCRIPTION

The genus *Rusa* is native to South and Southeast Asia, inhabiting diverse habitats ranging from dense forests to grasslands [1]. The Visayan Spotted Deer (VSD), also known as the Philippine Spotted Deer and *Rusa alfredi* (NCBI:txid1088129), is one of three endemic species in the Philippines and is a highly rare and endangered species indigenous to the Philippines' Visayan Islands. This region is considered one of the country's highest conservation priority areas, particularly due to the number of threatened endemic taxa and the degree of threats to species and habitats. Characterized by their soft dark-brown coat and unique nominal spots, *R. alfredi* once played a vital role as herbivores in shaping vegetation dynamics. However, its extirpation from most areas makes it difficult to determine its historical ecological impact fully. It has been classified as endangered since

1988 by the Red List of Endangered Species of the International Union for Conservation of Nature (IUCN). As of 2016, only an estimated 700 mature individuals remained in the wild. The genus *Rusa* is facing a significant decline in biodiversity worldwide and is under immense threat of global extinction.

The geographic range of *R. alfredi* formerly encompassed the Central Visayan islands of Negros, Panay, Guimaras, Masbate, and Cebu. Presently, only the islands of Panay and Negros shelter small, remnant populations of wild *R. alfredi* (Figure 1A) [2]. Accurate reports of the population density and distribution of the species in the wild have not yet been established. Like other cervid species in the world, the steep decline in the population of *R. alfredi* is mainly due to deforestation and hunting, despite being legally protected. Efforts to conserve the population of *R. alfredi* have been put in place, including the proposed creation of new national parks and properly structured captive breeding for reintroduction to the wild. The first captive breeding program for *R. alfredi* in the country was established at the Department of Biology and the Center for Tropical Conservation Studies (CENTROP) of Silliman University, in Dumaguete City, Negros Oriental, Philippines from Negros Island stock [3]. Presently, it has the largest captive-bred stock of the species globally.

Recent advancements in genomic sequencing created the possibility of producing large-scale reference genomes, which may offer new insights into an organism's genetic diversity and architecture. This enables researchers to identify key genetic traits, track evolutionary changes, and develop strategies for conservation and breeding programs aimed at preserving biodiversity and enhancing desirable traits in various organisms. Whereas several genetic technologies are already accessible, few are being used to their full potential. The IUCN lists 15,521 animal species as threatened, and less than 3% of these species have genomic resources that can inform and aid conservation management [4]. Currently, there is no available reference genome for *R. alfredi* or the genus *Rusa*. The generation of a reference genome would give us a better understanding of the history, diversity, and demographics of this endangered Visayan-endemic deer, which is significant for the management of the captive population. In this study, the first draft genome assembly of *R. alfredi* was generated using Illumina short-read sequences, and could serve as a reference for gene prediction, taxonomy, evolution, landscape genetics, and conservation genomics.

## METHODS

### Sample collection

The sampling was conducted under the Department of Environmental and Natural Resources (DENR) Region VII Gratuitous Permit No. 2022-17. The sample was obtained from a member of the captive population at the CENTROP, Silliman University, Dumaguete City, Negros Oriental, Philippines. A male deer (Abraham; Figure 1B) was restrained using a net, and a piece of ear tissue was collected using an ear notcher, a standard tool for ear tagging in animals. Before release, wound spray was applied to the ear to prevent infection and allow faster healing. The tissue sample was cleaned with 95% ethanol, placed in a 1.5 mL microcentrifuge tube with absolute ethanol, and stored at −20 °C for future use.

### DNA extraction and quantification

Extraction was performed at Silliman University using the Wizard® SV Genomic DNA Purification System following the manufacturer's protocol (Promega, 2012). The quality of

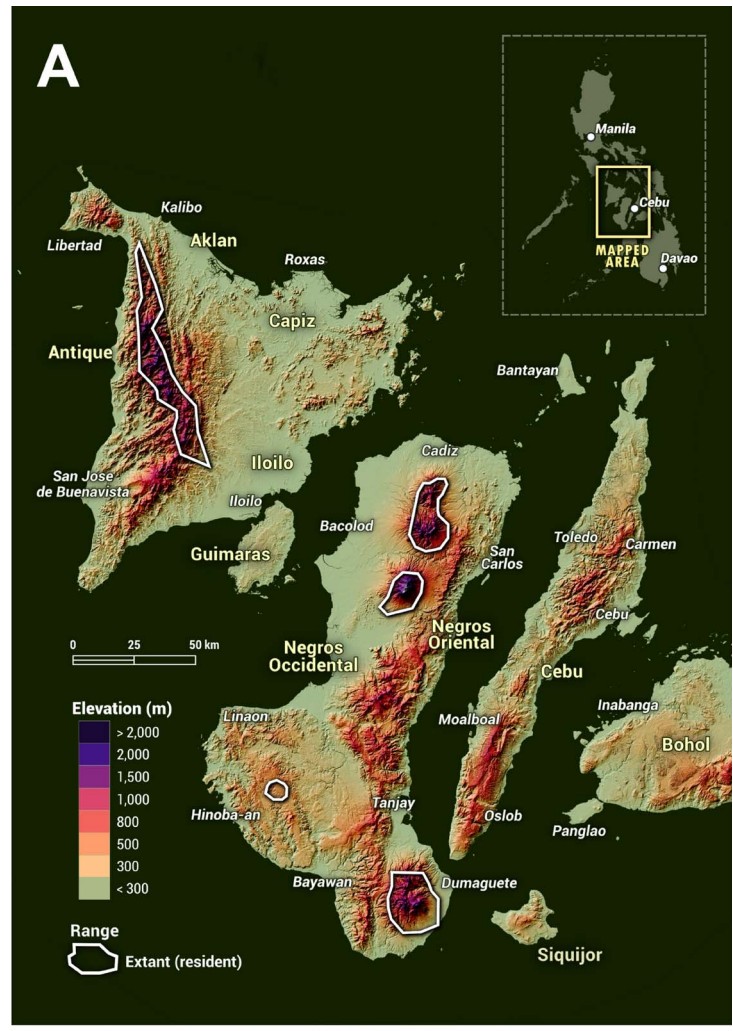

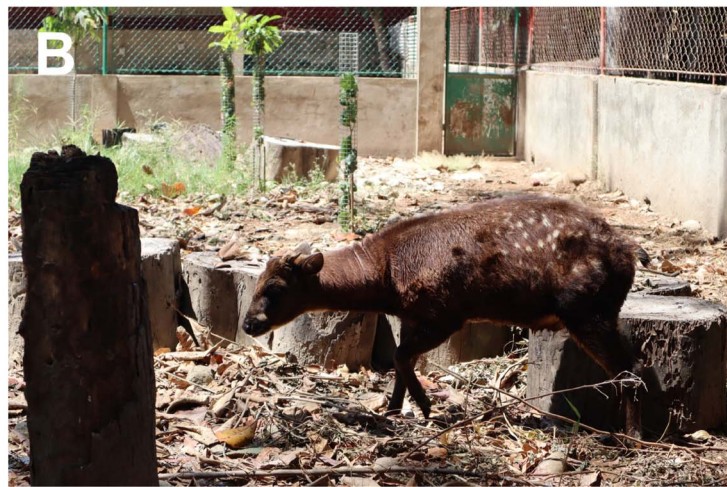

**Figure 1.** (A) Distribution Map of *R. alfredi* based on the IUCN Red List of Threatened Species [2]. (B) Photo of the Visayan Spotted Deer (code name: Abraham) at the CENTROP Silliman University, Dumaguete City, Negros Oriental, Philippines. Photo taken and shared by L. Cabrera, CC-BY.

the genomic DNA was subsequently checked using gel electrophoresis, Multiskan SkyHigh Spectrophotometer, and Qubit Fluorometer.

### Library preparation

The library construction was carried out with 100 ng of genomic DNA following the Illumina DNA library preparation kit manufacturer's protocol (Illumina, 2020). The resulting amplified library was quantified and controlled on an Agilent Bioanalyzer 2100 (Agilent, Santa Clara, CA) and sequenced in 2 × 151 bp paired-end reads on an Illumina NextSeq 1000 at the Philippine Genome Center Visayas, University of the Philippines Visayas, Miagao, Iloilo. A total of 157.47 Gbp of raw data was generated after sequencing.

### Genome survey

The quality of the short reads was checked using FastQC v0.12.1 (RRID:SCR_014583) [5]. To remove low-quality reads and sequencing adapters, reads were trimmed using Trimmomatic v0.39 (RRID:SCR_011848) [6] with the following parameters: ILLUMINACLIP: Nextera-PE-PE.fa:2:30:10 LEADING:30 TRAILING:30 SLIDINGWINDOW:4:20 MINLEN:36. The genome size of *Rusa alfredi* was estimated using a k-mer-based approach. K-mer frequencies were obtained using jellyfish (RRID:SCR_005491) [7]  and the command: jellyfish count -C -m 21 -s 1G <(zcat forwards_reads.fastq.gz) <(zcat reverse_reads.fastq.gz) -t 30. K-mer count histogram was then generated by running: jellyfish histo -t 10 mer_counts.jf > reads.histo. The resulting k-mer histogram was used in GenomeScope2 (RRID:SCR_017014) [8] to estimate the genome size and heterozygosity. GenomeScope2 was run using the command: genomescope.R -i reads.histo -o genomescope_21 -k 21.

### Genome assembly and quality assessment

Using the trimmed reads, the *R. alfredi* genome was assembled using MaSuRCA v4.1.0 (RRID:SCR_010691) [9]. The configuration file used for running MaSuRCA included "PE = pe 500 100" as the recommended safe insert size and standard deviation values for short reads and "GRAPH_KMER_SIZE = auto" for automatic selection of k-mer size ($k$ = 99 was selected). The MaSuRCA assembly pipeline was run using the command: "masurca config.txt". The configuration file was uploaded to GigaDB [10].

To improve the quality and contiguity of the assembly, contigs were corrected for misassemblies and scaffolded based on sequence homology using RagTag version 2.1.0 [11] with the *Cervus elaphus* genome (GenBank assembly accession number: GCF_910594005.1) as reference. Assembly correction was performed using RagTag with default parameters. Corrected contigs were then used for scaffolding using RagTag with default parameters.

General metrics for assessing the quality of the assembly were determined using QUAST v5.2.0 (RRID:SCR_001228) [12]. QUAST was run with the "*–large*" option and with the inclusion of the paired-end reads by adding the "*-1*" and "*-2*" flags to provide results for the assembly coverage. The contigs and scaffolds were also checked for completeness using Benchmarking Universal Single-Copy Orthologs, BUSCO v5.4.4 (RRID:SCR_015008) [13] using the cetartiodactyla_odb10. The assembled genome was visualized using Blobtoolkit v4.3.5 (RRID:SCR_025882) [14].

The quality of the assembly was evaluated using Merqury (RRID:SCR_022964) [15]. K-mer count from reads was obtained using the command: meryl k=21 *fastq.gz output

reads.meryl threads=30 memory=30. Assembly consensus quality value (QV), k-mer completeness, and spectra-cn plots were generated running the command:

merqury.sh reads.meryl GCA_038501445.1_RusAlf_1.1_genomic.fna abraham_merq.

The number of heterozygous sites and base coverage were determined based on the reads' alignment to the assembled genome. Reads were mapped back to the assembly using BWA-MEM (RRID:SCR_010910) [16] with the command: bwa mem -t 12 GCA_038501445.1_RusAlf_1.1_genomic.fna forward_reads.fastq.gz reverse_reads.fastq.gz | gzip -3 > aln-pe.sam.gz. Alignment was further processed using SAMtools v1.20 (RRID:SCR_002105) [17]. Specifically, mate information was added in the alignment using samtools fixmate, followed by samtools sort, and samtools markdup for marking and removing the duplicates (with -r flag). Average base coverage was determined using samtools depth. Alignment was also used for obtaining the raw variant call format (VCF) file using BCFtools v1.21 (RRID:SCR_005227) [17] by running the command: bcftools mpileup -Ou -f GCA_038501445.1_RusAlf_1.1_genomic.fna alignment.fxm.sorted.rmdup.bam | bcftools call -mv -Ov -o raw_variants.vcf. The filtered VCF file was obtained by running the command: bcftools filter -e 'QUAL < 30 || DP < 10' -o filtered_variant.vcf -O v raw_variants.vcf. The number of heterozygous sites was determined using the command: bcftools view -i 'GT="0/1"' filtered_variant.vcf | grep -v "ˆ#" | wc -l.

## Repeats and gene annotation

Before the annotation, the assembly was screened for contaminants and the presence of mitochondrial sequences. Detected mitochondrial sequences in the assembly were either trimmed or removed from the assembly using SeqKit v2.7.0 (RRID:SCR_018926) [18]. *De novo* identification of the repeats was performed in the assembly using RepeatModeler v2.0.5 (RRID:SCR_015027) [19]. The Database for RepeatModeler was first generated by running the command: BuildDatabase -name VSD GCA_038501445.1_RusAlf_1.1_genomic.fna. It was followed by *de novo* repeat identification using the command: RepeatModeler -database VSD -threads 12 -LTRStruct. The resulting library of repeats was then merged with the mammals repeat library extracted from the Dfam database [20] using *famdb.py* script. The mammalian repeat library was obtained using the command: famdb.py -i Dfam.h5 families -a -d -f fasta_name "mammals" > mammals_repeat_library.fasta. The combined libraries were then used to soft mask the repeats in the genome using RepeatMasker v4.1.5 (RRID:SCR_012954) [21] with '*-s -xsmall*' options. For gene annotation, homology-based gene prediction was performed using the Gene Model Mapper (GeMoMa v1.9, RRID:SCR_017646) Pipeline [22] with *Cervus elaphus* genome (GenBank accession number: GCF_910594005.1) as reference. GeMoMa was run using the command: GeMoMa -Xmx50G GeMoMaPipeline threads=12 outdir=GeMoMa GeMoMa.Score=ReAlign AnnotationFinalizer.r=NO o=true t=RusAlf_v1.1.fna a=mCerela.gff g=GCF_910594005.1_mCerEla1.1_genomic.fna. Additional gene annotation was obtained using the BRAKER v3.0.8 annotation pipeline C (RRID:SCR_018964) [22–33]. Vertebrata protein sequences from the OrthoDB v11 (RRID:SCR_011980) [34] partition were used to serve as extrinsic evidence for gene prediction in the soft-masked genome. BRAKER-annotated genes were filtered by retaining only those with hits in the Pfam database [35], identified using InterProScan v5.72-103 (RRID:SCR_005829) [36]. Verified annotated genes from BRAKER were then added to gene annotation from GeMoMa using AGAT v1.4.2 [37]. To ensure that gene annotation structures were retained, only gene annotations from BRAKER with no overlapping contained coding

sequences (CDS) were added to the gene annotations from GeMoMa to generate the final gene set using the agat_sp_complement_annotations.pl script. Protein sequences from the final gene set of *R. alfredi* were extracted for further downstream analysis.

### Phylogenetic tree

A phylogenetic tree of *R. alfredi* and other species of cervids was constructed based on single-copy orthologs. Protein sequences from reference genomes of *Odocoileus virginianus, Rangifer tarandus, Muntiacus muntjak, Muntiacus reevesi, Dama dama, Cervus hanglu yarkandensis, C. elaphus*, and *Cervus canadensis* were downloaded from GenBank, while sequences for *Cervus nippon* were downloaded from the Figshare database [38]. These sequences were used together with the predicted protein sequences of *R. alfredi* to create a species tree. Asian water buffalo (*Bubalus bubalis*) was included to serve as an outgroup. The longest transcript per gene in each species protein dataset was identified and retained using primary_transcript.py from OrthoFinder v.2.5.5 [39]. Single-copy orthologs were identified using OrthoFinder v.2.5.5 (RRID:SCR_017118) [39]. The sequences were renamed with the corresponding species ID using Seqkit v2.7.0 [18], and each ortholog was aligned using MUSCLE v5.1.0 (RRID:SCR_011812) [40]. Aligned sequences were then concatenated using Seqkit v2.7.0 [18], and trimming was performed using Gblocks v0.91b (RRID:SCR_015945) [41] with default parameters. The maximum likelihood tree was generated using IQ-TREE v2.3.6 (RRID:SCR_017254) [42] with ModelFinder [43] for model selection based on Bayesian Information Criterion (BIC) and bootstrap set at 1000. The maximum likelihood (ML) tree based on single-copy orthologs was constructed using the command: iqtree -s MSA_cervid_sco_concat_sorted_trimmed.fasta -m MFP -B 1000. The resulting ML tree was then visualized using iTOL (RRID:SCR_018174) [44].

### Mitochondrial genome assembly, annotation, and phylogenetics

The *R. alfredi* mitochondrial genome was also assembled using MITObim v1.9 (RRID:SCR_015056) [45]. The complete cytochrome oxidase I (COI) sequence from the existing *Rusa alfredi* complete mitogenome (NCBI Accession number JN632698.1) was used as seed fasta for the assembly. A random sampling of 20% of reads was performed using the following command: "downsample.py -s 20 –interleave -r forward_read -r reverse_read | gzip > sampled_20.fastq.gz". Sampled reads were then used for the assembly by running the command: MITObim.pl -start 1 -end 100 -sample mysample -ref myref -readpool sampled_20.fastq.gz -quick seed.fasta –pair. The circular topology of the assembly was checked using the command: circules.py -f assembled_mtDNA.fasta.

Annotation was then performed in the assembled mitogenome using MitoZ v3.6 [46]. Annotation of the assembled mitogenome was performed using the command: mitoz annotate –fastafiles Abraham_mtDNA_genome.fasta –outprefix annotation –thread_number 12 –clade Chordata. An ML tree was also constructed based on 13 concatenated coding sequences of mitochondrial genomes of different species of cervids. Concatenated coding sequences were aligned in MEGA11 (RRID:SCR_000667) [47] using MUSCLE [40]. After alignment, the ML tree was constructed using IQ-TREE v2.3.6 [42] with the use of ModelFinder [43] for model selection based on the BIC. The ML tree was constructed with a bootstrap of 1,000. Water buffalo (*B. taurus*) was selected as an outgroup. The phylogenetic tree was visualized in iTOL [44]. An ML tree based on single-copy orthologs was constructed using the command: iqtree -s MSA_cervid_sco_concat_sorted_trimmed.fasta -m MFP -B 1000. The resulting ML tree was then visualized using iTOL [44].

## RESULTS AND DISCUSSION

Reference genomes play a crucial role in understanding genetic variation and the molecular underpinnings of traits across various organisms. They facilitate gene annotation, regulatory elements identification, and the elucidation of biological processes. Molecular investigations in cervid species have predominantly focused on systematic relationships using mitochondrial genomes [48], leaving gaps in understanding the adaptive potential and genetic basis of traits and the resolution of deeper nodes (above the family level) in population studies. Moreover, mitochondrial genomes alone may not provide a complete reconstruction of a species' evolutionary history since it is maternally inherited. Furthermore, several species are underrepresented in genomic databases due to their threatened conservation status or lack of available data, hindering sample collection [49]. This study presents the draft genome assembly of *Rusa alfredi* (RusAlf_1.1), marking the first genome assembly for the genus *Rusa*. This contribution is pivotal for conducting integrative analyses essential for the conservation and management strategies of *R. alfredi* amidst the threats of human, environmental, and emerging diseases.

### Genome survey

The genome of *R. alfredi* (codename: Abraham) was estimated to be 2.37 GB in length with a low level of heterozygosity (0.30%) based on k-mer analysis using GenomeScope [8] (Figure 2). Based on the analysis using the mapped reads, a total of 4,305,197 (0.17%) heterozygous sites were identified, confirming the low heterozygosity of the genome. The genome size was also similar to the one estimated by MaSuRCA (2.37 GB) and close to the actual total length of the assembled contigs (2.51 GB). The K-mer distribution showed a single peak, indicating a high homozygosity (99.70%) of the assembled genome.

Captive populations have low heterozygosity compared to wild populations primarily due to factors like inbreeding and bottleneck effect [50], which limit genetic diversity in smaller, isolated groups. The variations between the two haplotypes in genomes with low heterozygosity often involve smaller-scale differences, making alignment easier during genome assembly and leading to accurate consensus sequences. In deer genomes, such as those of the sika deer [51] and the white-tailed deer [52], low levels of heterozygosity have been shown to simplify *de novo* assembly and improve alignment accuracy.

### Genome assembly and quality assessment

The assembled draft genome of *R. alfredi*, RusAlf_1.1, has 171,678 total contigs with a total length of 2.5 GB. The genome size of the assembled genome was found to be comparable to the genomes of other cervid species, such as *Cervus hanglu yarkadensis* with 2.6 GB (CEY_v1, GenBank accession GCA_010411085.1) and *Muntiacus muntjak* with 2.57 GB (UCB_Mmun_1.0, GenBank accession GCA_008782695.1). The assembled genome has short contiguity with N50 of 46 kb, which was expected considering the limitations of short paired-end reads (2 × 151 bp) to resolve the repeats in large genomes [53]. Long-read sequences are usually added for the assembly to achieve longer contiguity, which adds to the overall cost associated with genome assembly efforts. Genome assembly can be improved using reference genomes, provided the reference genome is closely related to the target species. The genus *Cervus* is one of the closest relatives of *R. alfredi* based on the phylogenetic tree of mitochondrial genomes of the tribe Cervini [54]. For the draft genome RusAlf_1.1, mCerEla1.1 (GenBank accession GCA_910594005.1) was used to correct

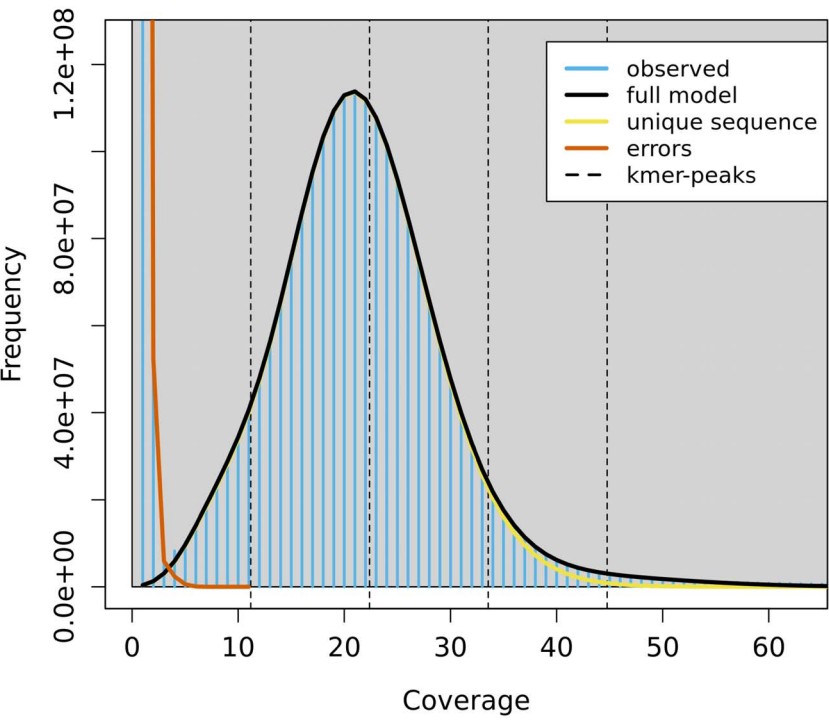

**Figure 2.** K-mer (21) distribution. GenomeScope2 was used to estimate the genome size and heterozygosity of the *Rusa alfredi* genome. len - estimated haploid genome length; aa - homozygosity; ab - heterozygosity; k-cov - mean heterozygous k-mer coverage, err - read error rate; dup - the average rate of read duplications; k: k-mer size used for the run; p - ploidy.

misassembled contigs based on sequence homology and improve the assembly through homology-based scaffolding. The final assembly has a total of 57,916 scaffolds, scaffold N50 of 75 MB, and scaffold L50 of 13 (Figure 3A). The same homology-based assembly was performed using *C. elaphus*, mCerEla1.1, as a reference for mounting contigs for chromosome-level assembly of the fallow deer (*Dama dama*) reference genome [55].

The quality of genome assemblies is generally assessed based on contiguity and completeness. It was highlighted that interpreting the quality of the assembly using metrics like N50 or L50 alone can be misleading, as it only measures the assembly contiguity and does not consider the assembly completeness and correctness [56]. In this study, despite the low level of contiguity, the draft genome of *R. alfredi* scored a high level of completeness with 95.5% complete BUSCO using cetartiodactyla_odb10 (*n* = 13,335). The assembled genome also scored a high Merqury QV of 47 (equivalent to about 99.99% base accuracy) and a completeness score of 96.76%. In addition, the k-mer spectrum plot shows a single high peak for 1-copy k-mer (red) and a very small area for 2-copy k-mer (blue), indicating a homozygous genome (Figure 4). The assembly quality was also checked by mapping the reads back to the final assembly using the QUAST pipeline. The read mapping results revealed that 99.38% of the reads were successfully mapped back to the assembly with a

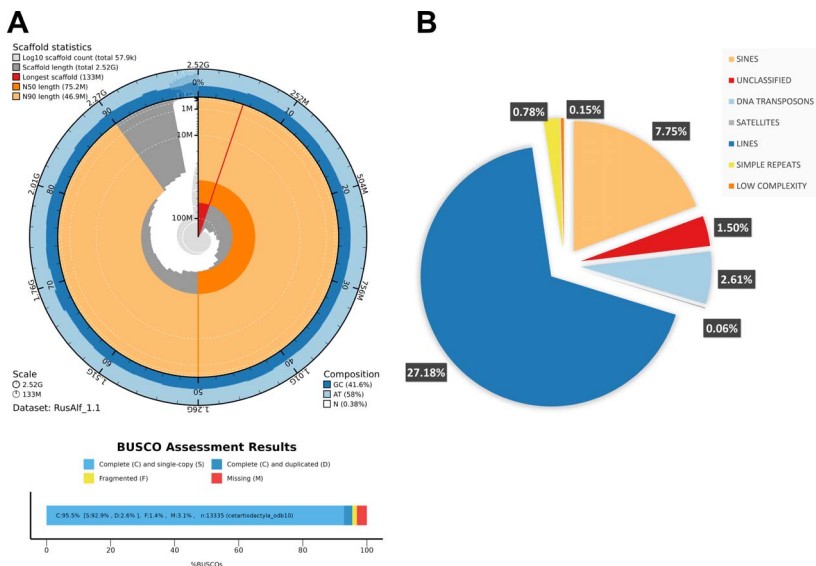

**Figure 3.** (A) Assembly metrics and BUSCO scores of RusAlf_1.1 and (B) Repeat elements in the draft genome of RusAlf_1.1.

mean base coverage of 47×. This study showed that a high level of assembly completeness of the draft genome can still be achieved using only short paired-end reads. It is worth noting that the estimated genome size from the genome survey is smaller than the assembled genome size mainly due to differences in methodology. K-mer analysis underestimates size by excluding repetitive sequences and errors, while a whole genome assembly includes all data, including repetitive regions and possible duplications. This results in a larger assembled genome size compared to the survey estimate.

Comparison of the BUSCO results between contigs (from MaSuRCA) and scaffolds (MaSuRCA+RagTag correct and scaffold) of the assembly shows improvement in the completeness of the assembly with a complete score increasing from 74.2% to 95.5% (Table 1). RagTag statistics after scaffolding also showed high confidence scores (average grouping confidence: 99.78%; average location confidence: 99.65%). However, it should be noted that using reference genomes of different species for scaffolding could introduce errors, considering the structural variations even between genomes of two related species. As there are no current genetic maps and limited related genomic resources for *Rusa alfredi*, structural variations in the genome could be addressed and validated in future studies by incorporating long-read sequencing as well as Hi-C libraries. Nevertheless, the current draft genome of *R. alfredi* serves as a valuable foundational resource for the continued conservation of this species.

## Genome annotation report

The RusAlf_1.1 genome is comprised of 44.27% of total interspersed repeat sequences. Most repeats were classified as retroelements, comprising 40.16% of the genome, followed by DNA transposons, 2.61% of the genome, and unclassified repeats, 1.50% of the genome (Figure 3B). The repetitive sequence analysis revealed similarities to several cervid species' genomes in terms of genomic composition. For instance, in the Sika deer (*Cervus nippon*),



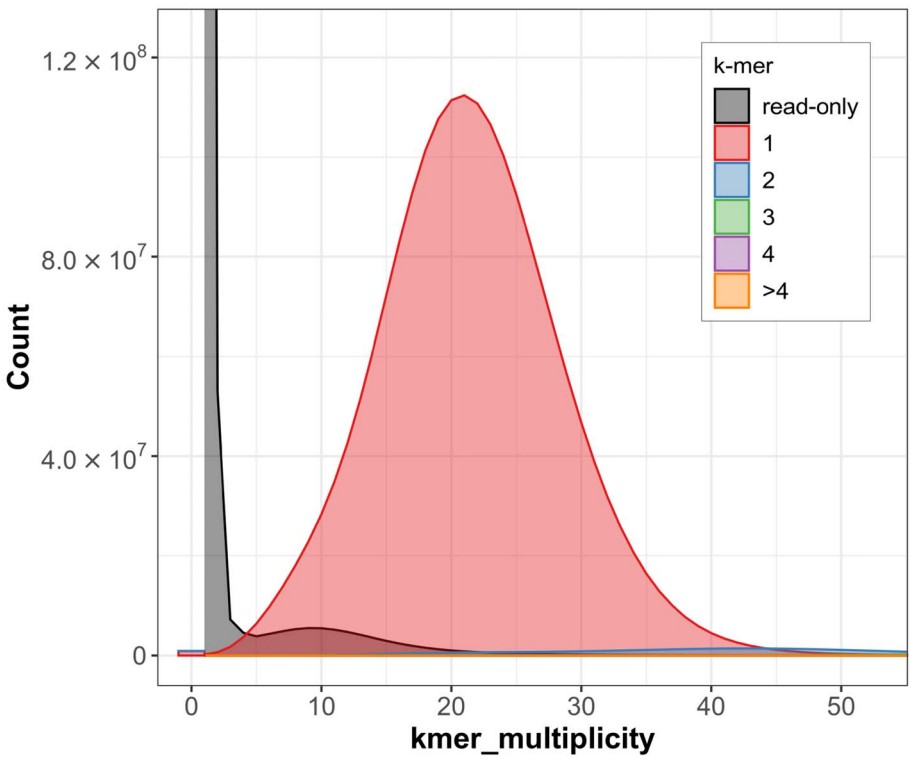

**Figure 4.** Merqury k-mer spectrum plot of the assembled genome of *Rusa alfredi* against the Illumina short paired-end reads. Read-only (grey) represents k-mers that are only found in reads but not in the assembly. Colors represent k-mers found in reads and the assembly 1× (red), 2× (blue), 3× (green), 4× (purple), and >4× (yellow).

**Table 1.** BUSCO summary results for contigs and scaffolds of the *Rusa alfredi* draft genome.

| BUSCO (*n* = 13,335, cetartiodactyla_odb10) | INITIAL CONTIGS (using MaSuRCA v4.1.0) | SCAFFOLD (with homology-based correction and scaffolding using RagTag v2.1.0) |
|---|---|---|
| **Complete (single + duplicated)** | 74.2% (9,901) | 95.5% (12,741) |
| **Single-copy** | 71.5% (9,538) | 92.9% (12,392) |
| **Duplicated** | 2.7% (363) | 2.6% (349) |
| **Fragmented** | 7.0% (935) | 1.3% (178) |
| **Missing** | 18.8% (2,499) | 3.2% (416) |

repetitive sequences make up around 45.38% of its genome [57]. Among repetitive elements, long interspersed nuclear elements (LINEs), short interspersed nuclear elements (SINEs), and long terminal repeats (LTRs) are the most abundant. Similar patterns are observed in Tarim red deer (*Cervus elaphus yarkandensis)* [58], Siberian musk deer (*Moschus moschiferus*) [59], white-tailed deer (*Odocoileus virginianus*) [60], and reindeer (*Rangifer tarandus*) [61], where repetitive sequences account for significant portions of their genomes, ranging from 39.1% to 42.4%. It was also found that the simple sequence repeats (SSRs) make up about 0.76% of the RusAlf_1.1 genome. SSRs or microsatellites are highly polymorphic loci that can be used for conservation genetics to estimate genetic structure. Genetic diversity plays a crucial role in wildlife management and disease mitigation, as demonstrated by studies on wild pig populations in Texas and roe deer in Iberia, emphasizing the need to integrate genetic data into conservation strategies [62, 63].

The captive population of Silliman University would directly benefit from the assembled genome in assisting their current genetic diversity studies.

Gene annotation of RusAlf_1.1 was initially performed through homology-based gene prediction. The addition of transcriptome data has been shown to improve the accuracy of gene prediction [64], especially for *de novo* gene prediction. However, obtaining transcriptome data for critically endangered species like *R. alfredi* is challenging due to its limited population size and the ethical and logistic constraints of sampling. Additionally, obtaining a sample for RNA-Seq in this study was not possible due to limited financial resources. Nevertheless, a total of 22,862 genes were predicted from the RusAlf_1.1 genome through homology, which is comparable to the 22,941 predicted genes in the red deer (*Cervus elaphus*) genome [65]. To further predict genes present in the genome, additional gene prediction was performed using the BRAKER pipeline C, incorporating a protein database for external evidence in gene prediction. BRAKER initially predicted a total of 35,129 genes, of which 16,343 were verified using InterProScan with the Pfam database. Among these verified genes, 1,669 CDS were unique to BRAKER and did not overlap with the GeMoMa annotation. These genes with non-overlapping CDS were added to the GeMoMa annotation, resulting in a final gene count of 24,531 for *R. alfredi*. The predicted genes in RusAlf_1.1 were then used to study the phylogenetic relationship of *R. alfredi* with other cervids with sequenced genomes.

## Phylogenetic inference

A phylogenetic analysis was constructed based on single-copy orthologs of different species of deer (Figure 5). The resulting phylogenetic tree showed a monophyletic grouping of the four species of *Cervus*, namely *C. elaphus* (GenBank accession GCA_910594005.1), *C. hanglu yarkandensis* (CEY_v1, GenBank accession GCA_010411085.1), *C. nippon* (GenBank accession GCA_040085125.1), and *C. canadensis* (GenBank accession GCF_019320065.1). A similar tree was depicted in a previous study with the addition of RusAlf_1.1 from this study [66]. The species tree revealed a close relationship between RusAlf_1.1 and the genus *Cervus*. This result supports a previous study based on complete mitochondrial genomes, suggesting that the genus *Rusa* is sister to *Cervus* [67]. However, the phylogenetic position of *R. alfredi* relative to other species of *Rusa* could not be evaluated due to the absence of genome data for other *Rusa* species. The continued efforts for the genome assembly of *Rusa* species will be crucial for elucidating the evolutionary relationships between *Cervus* and *Rusa*.

## Mitochondrial genome assembly, annotation, and phylogenetics

The complete mitochondrial genome for RusAlf_1.1 was also assembled using short paired-end reads. The final assembly has a 16,356 bp total length. A total of 13 coding genes, 22 tRNA genes, and two rRNA genes were annotated in the assembled mitogenome. The assembly was uploaded in the GenBank with accession number PQ083075.

An ML tree of different species of cervids using concatenated coding sequences of mitogenomes (Figure 6) showed subdivisions between subfamilies of cervids: Capreolinae and Cervinae. The monophyletic grouping of *R. alfredi* Abraham (GenBank Accession number PQ083075.1) and the reference mitogenome for *R. alfredi* (GenBank Accession number JN632698.1) was also observed. Our ML tree result further supports the close relationship between *R. alfredi* and the genus *Cervus*.

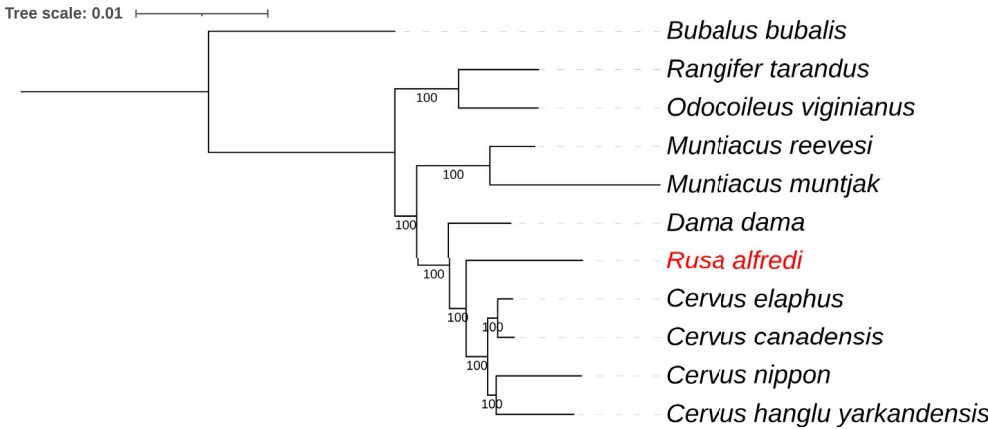

**Figure 5.** ML tree of different cervid species based on single-copy orthologs. The ML tree was constructed from the multiple sequence alignment (MSA) of 7,188 concatenated single-copy orthologs of nuclear genomes. The MSA reached a total of 3,992,528 amino acid sites after trimming. The ML tree was constructed using Q.mammal+F+I+R10 substitution model and bootstrap set at 1,000.

Mitogenome sequences have become valuable resources for elucidating phylogenetic relationships among different cervids. For instance, the proposed transfer of *Rucervus eldii* to the genus *Panolia* was due to mitogenomic evidence of its close relationship with *Elaphurus davidianus* and its separation from *Rucervus duvaucelii* [67, 68]. In this study, we found that *Rusa* forms an evolutionary grade with *Cervus* due to the position of the latter as a monophyletic clade nested within *Rusa. Rusa alfredi* was recovered as basal to the *Rusa + Cervus* clade, agreeing with a previous mitogenomic phylogeny [67]. The basal position of *R. alfredi* in the clade raises interesting questions about the evolutionary history of the cervids, particularly in island environments such as the Philippines. The patterns of speciation and diversification of cervids in insular southeast Asia require further study. It is recommended to also sequence the genome of *R. marianna*, another cervid endemic to the Philippines, as well as other allopatrically-distributed *Rusa* populations, to elucidate their evolutionary histories and taxonomic distinctiveness. Increased taxon sampling could also potentially serve to test the Pleistocene Aggregate Island Complex theory [69] by examining patterns of divergence, gene flow, and demographic history of deer populations that were potentially connected during periods of low sea levels but are currently separated in different islands. In addition, Pleistocene climate-driven changes in the availability of suitable habitats may have also caused disjunct distributions and diversification [70]. Paleoclimatic models can be incorporated to understand the physical and environmental factors that may have promoted diversification in Philippine cervids.

Previous studies showed evidence of hybridization between different species of *Cervus* [71, 72] and between species of *Rusa* [73]. Subsequent backcrossing of hybrids to the population could cause mitochondrial introgression, which could obscure or complicate phylogenetic reconstruction if based solely on the mitochondrial genomes. In the case of *R. alfredi*, although hybridization between *R. alfredi* and *R. marianna* was previously observed [2, 3], it was unlikely to happen in the current small population size and non-overlapping geographic distribution of the two species. Also, there is still a lack of genetic evidence for the previous report of hybridization between *R. alfredi* and *R. marianna* that can support the possible mitochondrial introgression in the current

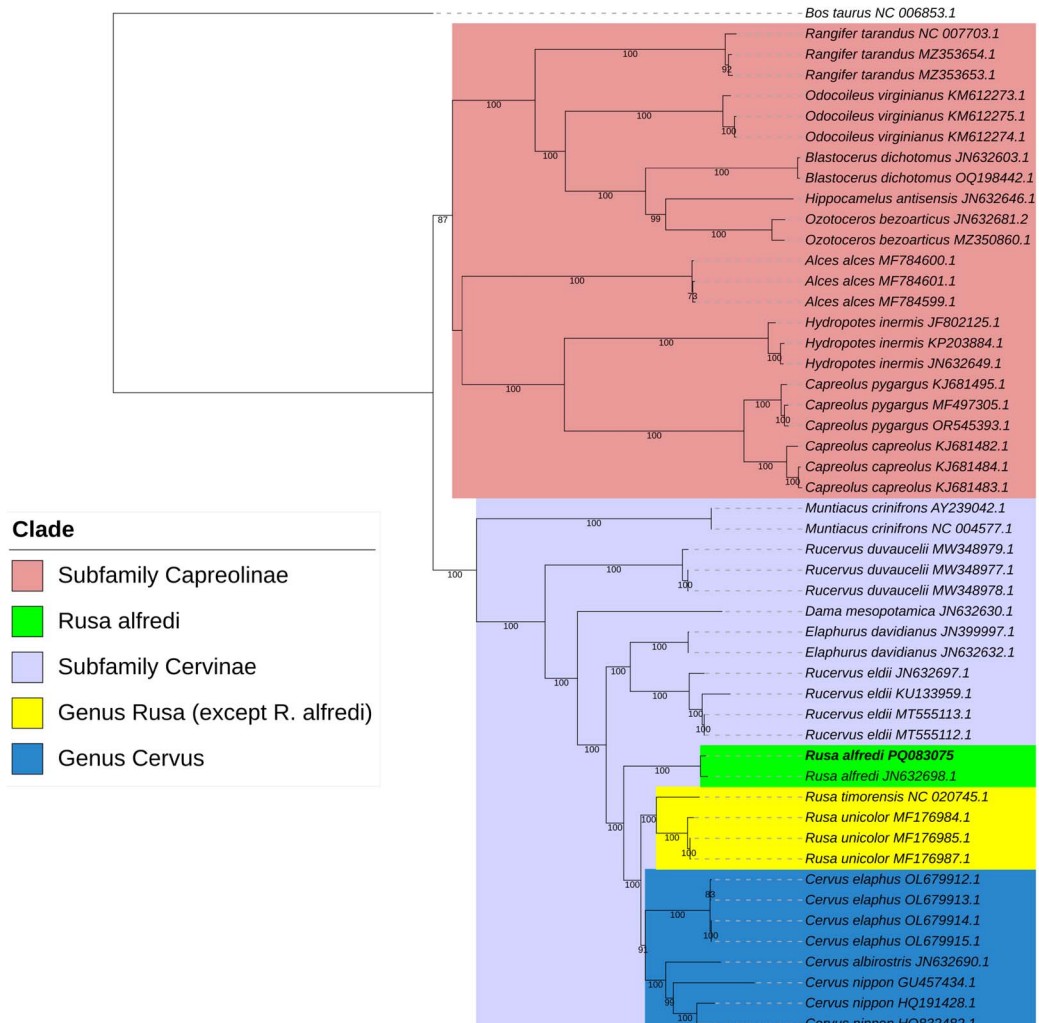

**Figure 6.** ML tree of different species of cervids based on concatenated coding sequences from complete mitochondrial genomes. The ML tree was constructed using the TIM2+F+I+R3 substitution model with 1,000 bootstrap replicates.

captive and wild populations of *R. alfredi*. Providing whole genome sequences for other native species of *Rusa* could further provide genomic resources for detecting hybrids, which will also help the management and monitoring of these species, especially for the reintroduction of captive populations in the wild.

The assembled genome of *R. alfredi* represents an advancement in the research and conservation efforts for this endangered endemic species. It not only reinforces previous taxonomic classifications of *R. alfredi* but also facilitates the evaluation of its evolutionary relationships with other species of *Rusa* and *Cervus* [3, 74]. This underscores the importance of obtaining additional genomic data from more *Rusa* and *Cervus* species. Considering the limitations of the draft assembly using short reads sequencing and the possibilities of misassembly given the used methods and resources, the quality of the

genome of *R. alfredi* can be improved by adding RNA-Seq, karyotyping to establish a clear chromosomal framework, integrating long-read sequencing to enhance contiguity and accuracy, and utilizing Hi-C libraries to detect and resolve structural variations. These approaches will not only refine the genome assembly but also provide critical insights into structural differences between *R. alfredi* and other *Cervus* species, ultimately contributing to more robust conservation strategies. Nevertheless, the initial availability of a genomic resource will support the development of targeted conservation strategies among the captive population. Incorporating samples from wild populations of *R. alfredi* will also allow us to identify genes that have evolved in captive settings, informing us about survival adaptations crucial for reintroduction efforts into the wild. This work enables further studies, such as microsatellite analysis, SNPs, RADseq, reference gene characterization, and whole-genome resequencing [75].

## DATA AVAILABILITY

The genome assembly generated in this study has been deposited at NCBI GenBank under the accession JBCEYX000000000. All sequencing reads can be accessed through the NCBI SRA (BioProject number: PRJNA1102104). Files generated in this study (Illumina reads, codes, configuration file for assembly, assembled genome, annotations, MSA, and phylogenetic tree files) are available in GigaDB [10].

## ABBREVIATIONS

BIC, Bayesian Information Criterion; CDS, contained coding sequences; CENTROP, Center for Tropical Conservation Studies; COI, cytochrome oxidase I; DENR, Department of Environmental and Natural Resources; IUCN, International Union for Conservation of Nature; LINEs, long interspersed nuclear elements; LTRs, long terminal repeats; ML, maximum likelihood; MSA, multiple sequence alignment; QV, quality value; SINEs, short interspersed nuclear elements; SSrs, simple sequence repeats; VCF, variant call format; VSD, Visayan Spotted Deer.

## DECLARATIONS

### Ethics approval and consent to participate

All study procedures and utility of experimental animals were conducted following the Republic Act No. 8485 or The Animal Welfare Act of 1998 of the Philippines. The tissue sampling carried out for this study was approved by the DENR Region VII (Gratuitous Permit No. 2022-17 Series of 2022). The authors declare that ethical approval was not required for this type of research.

### Competing interests

The authors declare that there is no conflict of interest.

### Authors' contributions

MCM, VMEF, CDC, RSG, and NPA conceptualized and supervised the study. VMEF secured the funding for the conduct of the study. PMS facilitated the permit for sample collection and handled sample preparation before sequencing. MCJ performed the experiment and managed the project. AN conducted the assembly and bioinformatics analysis. MCJ and AN wrote the manuscript with contributions from all authors. All authors reviewed and approved the final manuscript.

## Funding

This work was supported by the Philippine Genome Center Visayas, University of the Philippines Visayas.

## Acknowledgements

The authors would like to express their gratitude to Ozzy Boy S. Nicopior for his assistance in generating the distribution map used in this paper.

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
