## [Editor Report]

Editor’s AssessmentThe Visayan spotted deer (Rusa alfredi), is a small, endangered, primarily nocturnal species of deer found in the rainforests of the Visayan Islands in the Philippines. The present study reports the first draft genome assembly for the species, addressing a critical gap in genomic data for this IUCN-redlisted cervid. Using Illumina sequencing, the resulting genome assembly spans 2.52 Gb in size with a BUSCO completeness score of 95.5% and encompasses 24,531 annotated genes. Phylogenetic analysis suggests a close evolutionary relationship between R. alfredi and Cervus species suggesting that the genus Rusa is sister to Cervus. Peer-review teased out more benchmarking results and the annotation files, demonstrating this genomic resource is useful and usable for advancing population genetics and evolutionary studies, thereby informing conservation strategies and enhancing breeding programs for the critically threatened species. Providing whole genome sequences for other native species of Rusa could further provide genomic resources for detecting hybrids, which will also help the management and monitoring of these species, especially for the reintroduction of captive populations in the wild.Editor’s AssessmentThe Visayan spotted deer (Rusa alfredi), is a small, endangered, primarily nocturnal species of deer found in the rainforests of the Visayan Islands in the Philippines. The present study reports the first draft genome assembly for the species, addressing a critical gap in genomic data for this IUCN-redlisted cervid. Using Illumina sequencing, the resulting genome assembly spans 2.52 Gb in size with a BUSCO completeness score of 95.5% and encompasses 24,531 annotated genes. Phylogenetic analysis suggests a close evolutionary relationship between R. alfredi and Cervus species suggesting that the genus Rusa is sister to Cervus. Peer-review teased out more benchmarking results and the annotation files, demonstrating this genomic resource is useful and usable for advancing population genetics and evolutionary studies, thereby informing conservation strategies and enhancing breeding programs for the critically threatened species. Providing whole genome sequences for other native species of Rusa could further provide genomic resources for detecting hybrids, which will also help the management and monitoring of these species, especially for the reintroduction of captive populations in the wild.

---

## [Reviewer Report]

Indicate in the comments box below whether you are happy with the changes made or if the manuscript is unacceptable.Comments on revised manuscriptI thank the authors for their efforts to address the concerns raised. I broadly agree with the answers, but three further details need clarification: 1. Calculating the raw reads and the resulting genome size yields a coverage of about 62x. The authors mapped the raw reads back to the resulting reference genome sequence, which gave 47x coverage. However, both Genomescope and Merqury K-mer analysis showed 22x coverage. What is the reason for this discrepancy? 2. The K-mer analysis does indeed, and a bit strangely, show what appears to be a haploid genome. However, the 0.302% heterozygosity measured by GenomeScope is not remarkably low. To have an accurate picture of this, it would be important to count the number of heterozygous sites based on the raw reads mapped back at 47x coverage. 3. Although we do not know the exact chromosome number, fitting the reference to the red deer reference could be interesting. It would show how many scaffolds fit more than one red deer chromosome. Of course, this could be either due to chromosome rearrangement or because the contigs' scaffolding or assembly was incorrect.

---

## [Reviewer Report]

Indicate in the comments box below whether you are happy with the changes made or if the manuscript is unacceptable.Comments on revised manuscriptQ1：Why is the estimated genome size from the genome survey much smaller than the assembled genome size? Q2:In the method section, I did not see a description of the de novo method for gene structure annotation. Q3:I am concerned about using a reference genome with unclear karyotype relationships for scaffolding. Q4:Are there other published comparative genomic studies on deer that have identified such a small number of homologous genes?

---

## [Reviewer Report]

Reviewer name and names of any other individual's who aided in reviewer Endre BartaDo you understand and agree to our policy of having open and named reviews, and having your review included with the published papers. (If no, please inform the editor that you cannot review this manuscript.)YesIs the language of sufficient quality?YesPlease add additional comments on language quality to clarify if needed
Are all data available and do they match the descriptions in the paper? NoAdditional CommentsThe authors provided only the assembly in Fasta and GenBank format and the contigs (scaffolds?) in GenBank format. Neither the annotation nor the raw Illumina reads are available.Are the data and metadata consistent with relevant minimum information or reporting standards? See GigaDB checklists for examples <a href="http://gigadb.org/site/guide" target="_blank">http://gigadb.org/site/guide</a>YesAdditional CommentsIn the cases where the data is uploaded, the provided metadata is consistent.Is the data acquisition clear, complete and methodologically sound?YesAdditional CommentsIs there sufficient detail in the methods and data-processing steps to allow reproduction?NoAdditional CommentsThe exact parameters used during the processing are completely missing. For example, it is unclear how the RagTag-based correcting and scaffolding were carried out.Is there sufficient data validation and statistical analyses of data quality? Not my area of expertiseAdditional CommentsIs the validation suitable for this type of data?NoAdditional CommentsWithout having the raw Illumina reads and the exact command line parameters used, it is not possible to validate the provided results.Is there sufficient information for others to reuse this dataset or integrate it with other data?YesAdditional CommentsAny Additional Overall Comments to the AuthorAssembling the reference genomes of endangered species is a task of immense importance, with the potential to significantly advance our understanding and conservation of these species. This work provides an initial genome assembly based on Illumina short-read sequencing. The correction and scaffolding of the contigs were made with the RagTag program using the red deer PacBio-based chromosome-level assembly. The potential benefits of this work are vast, from gaining knowledge to initiating and furthering population studies to preserve the species. According to the annotation and the BUSCO analysis, the final assembly seems especially good, considering that it is short-read based. However, there are some concerns about the methodology and the provided data. 1. The Illumina short reads and the annotation data (GFFs, VCFs) are not available. 2. The methods used are not reproducible because the descriptions of the exact parameters are missing. 3. It seems that the authors did not use the ‘-r’ parameter during the scaffolding, which resulted in inserting 100bp Ns instead of the actual size insertion based on the red deer reference genome. 4. There is no K-mer based genome size estimation. 5. The chromosome number is not known. Is there any chromosomal rearrangement between the red deer and the Visayan Spotted Deer? 6. It is not justified why the protein- and mitochondria-based trees are drawn as cladograms and not as phylograms. This way, the actual distances between the different species cannot be seen. 7. Although the short reads were mapped back to the assembly, no variation data is provided. 8. Is it necessary to include these high number (46104) short (1000>) contigs in the assembly? 9. Although the red deer assembly was used for the correction and scaffolding, the annotation was compared to the mule deer.RecommendationMajor Revision

---

## [Reviewer Report]

Reviewer name and names of any other individual's who aided in reviewer Haimeng LiDo you understand and agree to our policy of having open and named reviews, and having your review included with the published papers. (If no, please inform the editor that you cannot review this manuscript.)YesIs the language of sufficient quality?YesPlease add additional comments on language quality to clarify if needed
Are all data available and do they match the descriptions in the paper? NoAdditional CommentsThe genomic annotation file is not publicly available.Are the data and metadata consistent with relevant minimum information or reporting standards? See GigaDB checklists for examples <a href="http://gigadb.org/site/guide" target="_blank">http://gigadb.org/site/guide</a>NoAdditional CommentsGenomic annotation information and protein sequence information were not found in the NCBI database.Is the data acquisition clear, complete and methodologically sound?YesAdditional CommentsIs there sufficient detail in the methods and data-processing steps to allow reproduction?NoAdditional CommentsIs there sufficient data validation and statistical analyses of data quality? YesAdditional CommentsIs the validation suitable for this type of data?YesAdditional CommentsIs there sufficient information for others to reuse this dataset or integrate it with other data?NoAdditional CommentsAny Additional Overall Comments to the AuthorThe manuscript, 'Draft Genome of the Endangered Visayan Spotted Deer (Rusa alfredi), a Philippine Endemic Species,' contributes to the field of conservation genomics. The study presents the first draft genome assembly of the Visayan Spotted Deer, utilizing Illumina short-read sequencing technology to generate valuable genomic resources for this endangered species. Here are some questions and comments. Q1. Why was gene annotation conducted using only homology-based annotation? It is recommended that the annotation approach includes de novo, RNA-based, and homology-based methods. Combining these approaches would provide a more comprehensive gene set, particularly for species with limited genomic resources. Please revise the method section to include these additional annotation strategies. The authors have stated that due to sampling limitations, RNA-based experiments could not be conducted. RNA extraction might be performed using the tissue samples that were previously collected for genome assembly. In Lines 167-172 Q2. Before proceeding with genome assembly, it is essential to conduct a genome survey. This initial step provides crucial information about the genome's size, complexity, and composition, which is vital for planning the assembly strategy and selecting appropriate sequencing technologies and bioinformatics tools. The survey should include estimates of genome size, GC content, repetitive elements, and ploidy level. Additionally, the result could be used to assess the completeness of the assembly. Please include a section on the genome survey in the Method section. Q3. To enhance the quality and contiguity of the assembly, utilizing another species as a reference genome for scaffolding might introduce errors due to discrepancies in karyotype. It is essential to ascertain whether there is a definitive karyotype study that verifies the consistency of the karyotype between the Visayan Spotted Deer and the reference species, indicating the absence of chromosomal fission or fusion events. In Lines 236-238 This information is crucial for the reliability of the scaffolding process. Q4. Although the length of scaffold N50 is long, the high number of scaffolds and contigs suggests fragmentation. Have you addressed redundancy in the assembly? In Line 238 Q5. Have you used software like Merqury to detect assembly errors and assess the completeness of the assembly? This is useful for evaluating the quality of the genome sequence and identifying potential issues that may need to be addressed. Q6. Are the species divergent, which might explain the low number of orthologous genes? Is this an annotation issue or does it reflect true biological divergence? Further investigation into the annotation process and comparative genomic analyses may be warranted to understand the extent of divergence and the implications for the study. In Lines 313-317 Q7. Please standardize the format of numbers throughout the manuscript to maintain consistency in the number of significant figures. In Lines 224, 225, 227, 239, 245RecommendationMajor Revision